# Gellan Gum Promotes the Differentiation of Enterocytes from Human Induced Pluripotent Stem Cells

**DOI:** 10.3390/pharmaceutics12100951

**Published:** 2020-10-10

**Authors:** Shimeng Qiu, Tomoki Kabeya, Isamu Ogawa, Shiho Anno, Hisato Hayashi, Tatsuro Kanaki, Tadahiro Hashita, Takahiro Iwao, Tamihide Matsunaga

**Affiliations:** 1Department of Clinical Pharmacy, Graduate School of Pharmaceutical Sciences, Nagoya City University, 3-1 Tanabe-dori, Mizuho-ku, Nagoya 467-8603, Japan; kaeffie9325@hotmail.com (S.Q.); kabeyatomoki@gmail.com (T.K.); i.o.music.0136@gmail.com (I.O.); thashita@phar.nagoya-cu.ac.jp (T.H.); tiwao@phar.nagoya-cu.ac.jp (T.I.); 2Nissan Chemical Corporation, 5-1 Nihonbashi 2-chome, Chuo-ku, Tokyo 103-6119, Japan; annos@nissanchem.co.jp (S.A.); hayashihi@nissanchem.co.jp (H.H.); kanaki@nissanchem.co.jp (T.K.)

**Keywords:** human induced pluripotent stem cells, gellan gum, enterocytes, drug transporter, drug development

## Abstract

The evaluation of drug pharmacokinetics in the small intestine is critical for developing orally administered drugs. Caucasian colon adenocarcinoma (Caco-2) cells are employed to evaluate drug absorption in preclinical trials of drug development. However, the pharmacokinetic characteristics of Caco-2 cells are different from those of the normal human small intestine. Besides this, it is almost impossible to obtain primary human intestinal epithelial cells of the same batch. Therefore, human iPS cell-derived enterocytes (hiPSEs) with pharmacokinetic functions similar to human intestinal epithelial cells are expected to be useful for the evaluation of drug absorption. Previous studies have been limited to the use of cytokines and small molecules to generate hiPSEs. Dietary fibers play a critical role in maintaining intestinal physiology. We used gellan gum (GG), a soluble dietary fiber, to optimize hiPSE differentiation. hiPSEs cocultured with GG had significantly higher expression of small intestine- and pharmacokinetics-related genes and proteins. The activities of drug-metabolizing enzymes, such as cytochrome P450 2C19, and peptide transporter 1 were significantly increased in the GG treatment group compared to the control group. At the end point of differentiation, the percentage of senescent cells increased. Therefore, GG could improve the differentiation efficiency of human iPS cells to enterocytes and increase intestinal maturation by extending the life span of hiPSEs.

## 1. Introduction

During drug discovery and development, many candidate drugs fail to enter clinical trials despite the fact that a large amount of money and time has been spent [1,2]. To predict the effectiveness and safety of a novel drug that is absorbed and metabolized by the small intestine and liver [3,4,5], it is important to accurately evaluate absorption and metabolism in the small intestine. In preclinical trials of drug development, Caucasian colon adenocarcinoma (Caco-2) cells and intestinal microsomes are used to evaluate intestinal drug absorption and metabolism, respectively [6,7]. However, these systems have some disadvantages, possibly leading to drug development failures. Firstly, the expression patterns of drug transporters in Caco-2 cells are different from those in the normal human small intestine [8,9,10]. Secondly, it is almost impossible to obtain primary human intestinal epithelial cells of the same batch for pharmacokinetic studies [11]. Lastly, microsomes can only evaluate drug metabolism by the enzymes existing in the fraction. 

For in vitro assessments, the main challenge faced by many researchers is that the cells used should have functions similar to the physiological state of the human body. Human induced pluripotent stem (iPS) cells derived from mature cells can differentiate into various tissue cells. These cells have the same genome backgrounds as the host cell. Therefore, the use of iPS cell-derived hepatocytes and intestinal cells for drug development studies [12] could help overcome the stated challenges. To date, some attempts have been made to generate small intestinal cells from human iPS cells. Several studies have demonstrated that some small molecular compounds could improve the differentiation of human iPS cells into enterocytes [13,14,15,16]. However, the function of such enterocytes is poor when compared with the small intestine cells in the human body. Therefore, we focused on generating enterocytes with higher pharmacokinetic properties from human iPS cells.

Recently, some researchers have demonstrated that dietary fibers are useful for maintaining healthy intestinal functions and environment [17,18]. Reproducing or mimicking the in vivo environment could be critical for the growth of cells in vitro. GG is a water-soluble anionic polysaccharide that is widely used in tissue engineering, gene delivery, and the pharmaceutical industry [19,20,21,22]. It has been approved by the U.S. Food and Drug Administration (FDA) as a food additive. In this study, we chose gellan gum (GG), a type of dietary fiber, to clarify whether dietary fiber could improve the differentiation of human iPS cells into enterocytes. 

## 2. Materials and Methods 

### 2.1. Human iPS Cells

Human iPS cell line Windy was provided by Dr. Akihiro Umezawa, National Center for Child Health and Development. Cells were cultured on mitomycin C-treated mouse embryonic fibroblasts in a 1:1 mixture of Dulbecco’s modified Eagle’s medium (DMEM) and Ham’s nutrient mixture F-12 (DMEM/F-12) (Wako Pure Chemical Industries, Osaka, Japan) basal medium with 20% KnockOut Serum Replacement (Thermo Fisher Scientific, Carlsbad, CA, USA), 2 mM l-glutamine (Wako Pure Chemical Industries), 1% minimum essential medium nonessential amino acid solution (NEAA) (Wako Pure Chemical Industries), 0.1 mM 2-mercaptoethanol (β-MeE) (Sigma-Aldrich, St. Louis, MO, USA), and 5 ng/mL basic fibroblast growth factor (FGF2) (PeproTech Inc., Rocky Hill, NJ, USA).

### 2.2. Human iPS Cell-Derived Enterocyte Differentiation

The cells were differentiated as previously described [13]. Briefly, human iPS cells were differentiated using Roswell Park Memorial Institute (RPMI) + GlutaMAX medium (Thermo Fisher Scientific) containing 0.5% fetal bovine serum (FBS) (Sigma-Aldrich), 100 units/mL penicillin G, 100 µg/mL streptomycin sulfate (Cosmo Bio Co., Tokyo, Japan), and 100 ng/mL activin A (PeproTech Inc.) for 2 days. To induce differentiation into the definitive endoderm, the cells were cultured in the medium with 2% FBS for 1 day. To induce differentiation of the definitive endoderm into small intestinal stem cells, the cells were cultured in DMEM/F12 basal medium with 2% FBS, 1% GlutaMAX (Thermo Fisher Scientific), 100 units/mL penicillin G, 100 µg/mL streptomycin sulfate, and 250 ng/mL FGF2 for 4 days. The differentiated small intestinal stem cells were subcultured on growth factor reduced Matrigel (BD Biosciences, Bedford, MA, USA)-coated plates. After seeding, the medium was changed to an enterocyte differentiation medium containing DMEM/F12 medium with 2% FBS, 1% GlutaMAX, 0.1 mM NEAA, 100 units/mL penicillin G, 100 µg/mL streptomycin sulfate, 2% B27 (Thermo Fisher Scientific), 1% N2 (Thermo Fisher Scientific) supplement, and 20 ng/mL epidermal growth factor (EGF) (PeproTech Inc.) and was cultured for 7 days. On day 14 since differentiation, 20 µM 2-(2-amino-3-methoxyphenyl)-4*H*-1-benzopyran-4-one (PD98059) (Wako Pure Chemical Industries), 5 µM 5-aza-2′-deoxycytidine (5-aza-2′-dC) (Wako Pure Chemical Industries), and 0.5 µM 3-(6-Methyl-2-pyridinyl)-*N*-phenyl-4-(4-quinolinyl)-1*H*-pyrazole-1-carbothioamide (A-83-01) (Wako Pure Chemical Industries) were supplemented to induce differentiation. In addition, 0.01% GG (Nissan Chemical Corporation, Tokyo, Japan) was added from day 8 to the end of differentiation (Figure 1).

### 2.3. Real-Time RT-PCR

After the cells had differentiated, cellular RNA was extracted using the Agencort RNAdvance Tissue Kit (Beckman Coulter, Brea, CA, USA) according to the manufacturer’s instructions. We calculated the RNA concentration by measuring the absorbance at 260 nm using Thermo NanoDrop One (Thermo Fisher Scientific, Waltham, MA, USA). Thereafter, we synthesized high-quality cDNAs for real-time polymerase chain reaction (PCR) using the ReverTra Ace® qPCR RT Master Mix (Toyobo, Osaka, Japan) according to the manufacturer’s instructions. KAPA SYBR FAST qPCR Kit Master mix (2×) ABI Prism (Sigma-Aldrich) was used for real-time RT-PCR, with a final volume of 10 µL. PCR primer sequences are shown in Table 1. The samples were reacted using the LightCycler® 96 System (Roche Diagnostics, Basel, Switzerland). The relative expression levels of the target genes were calculated using the 2^−ΔΔCT^ method, and hypoxanthine phosphoribosyltransferase (HPRT) was used as the housekeeping gene. 

### 2.4. Immunofluorescence Staining

For villin 1 and fatty acid-binding protein 2 (FABP2) staining, the differentiated cells were fixed and permeabilized in methanol (−20 °C) (Wako Pure Chemical Industries). For peptide transporter 1 (PEPT1), breast cancer resistant protein (BCRP), sodium-glucose co-transporter 1 (SGLT1), P-glycoprotein (P-gp), and integrin α5 staining, the differentiated cells were fixed in 4% paraformaldehyde solution (Wako Pure Chemical Industries) and then permeabilized in 0.1% Triton X-100. The cells were then blocked in phosphate buffered saline (PBS) containing 5% FBS. Cells stained with P-gp were blocked at every wash step using PBS containing 0.1% bovine serum albumin (Thermo Fisher Scientific). The cells were incubated overnight at 4 °C with 1:50 anti-villin (Santa Cruz Biotechnology, Inc., Dallas, TX, USA), 1:200 anti-FABP2 (R&D Systems, Inc., Minneapolis, MN, USA), 1:50 anti-PEPT1 (Santa Cruz Biotechnology, Inc.), 1:100 anti-BCRP (Abcam, Cambridge, UK), 1:50 anti-SGLT1 (Novusbio Biologicals, Centennial, CO, USA), 1:25 anti-P-gp (Abcam), or 1:100 anti-integrin α5 (Abcam). The cells were incubated with a 1:200 dilution of Alexa Fluro 488- and 568-labeled secondary antibodies (Thermo Fisher Scientific) for 60 min at 25 °C. PBS containing 5% FBS was used for the dilution of primary and secondary antibodies. The cells were incubated with 0.2 µg/mL 4′,6-diamidino-2-phenylindole (Dojindo Molecular Technologies, Inc., Kumamoto, Japan) for 5 min. The PEPT1-, BCRP-, and SGLT1-stained cells were covered with PBS and viewed using the Operetta High-Content Imaging System (PerkinElmer, Waltham, MA, USA). The numbers of SGLT-1-, BCRP-, PEPT1-, and integrin α5-positive cells were also counted using this instrument. The integrin α5- and P-gp-stained cells were viewed using the LSM 510 Meta confocal microscope (Carl Zeiss Inc., Oberkochen, Germany) and LSM 800 with Airyscan confocal microscope (Carl Zeiss Inc.), respectively.

### 2.5. Uptake Assay

The culture medium was removed and the cells were preincubated with the transport buffer (Hanks’ balanced salt solution containing 10 mM 2-(*N*-morpholino)ethanesulfonic acid (MES), pH 6.0) at 37 °C for 15 min. The uptake assays were initiated by replacing the medium with transport buffer containing 135 nM [3H]glycylsarcosine at 37 °C in the presence or absence of 3 mM ibuprofen (Wako Pure Chemical Industries) (PEPT1 inhibitor) or at 4 °C. Uptake was stopped by the addition of ice-cold transport buffer; the cells were rinsed using the same buffer. The cells were lysed with 0.2 M NaOH solution (0.5 mL/well) containing 0.5% sodium dodecyl sulfate (SDS). Radioactivity was measured through liquid scintillation counting, using 3 mL Clear-sol I (Nacalai Tesque, Inc., Kyoto, Japan) as the scintillation fluid. The total protein in differentiated cells was measured using the Pierce™ BCA Protein Assay Kit (Thermo Fisher Scientific).

### 2.6. Drug-Metabolizing Enzyme Activities

Midazolam, diclofenac, and (S)-mephenytoin were used as the substrates for cytochrome P450 (CYP) 3A4/5, CYP2C9, and CYP2C19, respectively. The differentiated cells were washed with enterocyte differentiation medium and incubated in enterocyte differentiation medium containing these substrates in the presence or absence of 10 µM ketoconazole (CYP3A4/5 inhibitor) for 24 h at 37 °C. After incubation, the reaction was stopped by placing the cell-culture plates on ice and collecting the medium for metabolite measurements. The cells were lysed and the amount of protein was measured using the Pierce™ BCA Protein Assay Kit. Metabolic activities were corrected using protein amounts. Measurements of the metabolites (1-hydroxymidazolam, 4′-hydroxydiclofenac, and 4′-hydroxymephenytoin) were conducted using ultraperformance liquid chromatography–tandem mass spectrometry analysis.

### 2.7. Cell Cycle Analysis

The Propidium Iodide Flow Cytometry Kit (Abcam) was used for cell cycle analysis. The cells were harvested at days 14, 24, and 26 according to the manufacturer’s instructions. The collected cells were stored at 4 °C. At day 26, the samples were analyzed using BD FACSVerse (BD Biosciences).

### 2.8. β-Galactosidase Assay

β-galactosidase staining was performed at day 26 of differentiation using SPiDER-βGal (Dojindo Molecular Technologies, Inc.) according to the manufacturer’s instructions. All stained cells were covered with PBS and viewed under the Operetta High-Content Imaging System. The number of positive cells was counted and calculated using this instrument.

### 2.9. Western Blotting

The cells were collected using a cell lysis buffer containing 50 mM Tris (pH 6.8), 2% SDS, 10% glycerol, and 10% β-MeE. Proteins were separated via sodium dodecyl sulfate–polyacrylamide gel electrophoresis using Criterion™ TGX™ 12% (Bio-Rad Laboratories, Inc., Hercules, CA, USA). The separated proteins were transferred onto a polyvinylidene difluoride membrane, and the membrane was blocked with BlockAce for 60 min at room temperature. The membranes were incubated with anti-GAPDH (1:1000) and anti-integrin α5 (1:1000) antibodies overnight at 4 °C and with a peroxidase-conjugated secondary antibody (1:1000) for 60 min at room temperature. The protein bands were detected using the Pierce ECL Western blotting substrate (Thermo Fisher Scientific Inc.) for GAPDH and the SignalFire Elite ECL reagent (Cell Signaling Technology, Danvers, MA, USA) for integrin α5. The proteins were detected and analyzed using the Amersham Imager 600 (GE Healthcare, Chicago, IL, USA).

### 2.10. Statistical Analysis

Statistical significance was assessed using Student’s t-test for comparisons between the two groups or via analysis of variance followed by Dunnett’s test or Tukey’s test for multiple comparisons. 

## 3. Results

### 3.1. Effects of GG on iPS Cell-Derived Enterocyte Differentiation

To evaluate the optimal concentration of dietary fiber required for differentiation, we preliminarily investigated the effects of several dietary fibers, including crystalline cellulose (0.015%), chitin powder (0.015%), bamboo pulp (0.015%), and GG (0.015%), on differentiation (Figure 1). 

Enterocytes cocultured with crystalline cellulose showed the highest expression levels of PEPT1, villin 1, and CYP3A4. However, due to its high viscosity and low dispersion, crystalline cellulose tends to gelatinize at 37 °C; therefore, its experimental operation is complicated. In addition, lot-to-lot variations occur because of these properties. As a result, it is challenging to use crystalline cellulose in research. We chose GG to evaluate the effects of fiber on enterocyte differentiation because it is easy to use and showed the most stable expression profile of marker genes during differentiation across all fiber-treated groups (Appendix A). We confirmed that the increase in P-gp expression did not depend on the concentration of GG compared with the control. The expression levels of villin 1, intestine specific homeobox (ISX), and SGLT1 were significantly higher in the GG group (0.01%) than in the control group (0.015%) (Appendix A). Therefore, 0.01% GG concentration was determined as optimal and was used to evaluate the impact of GG on enterocyte differentiation. The expression levels of small intestinal-specific markers, including villin 1 [23,24], ISX [25], FABP2 [26], dipeptidyl peptidase-4 (DPP4) [27], and caudal-type homeobox 2 (CDX2) [25,28], and the tight junction marker occludin were significantly increased in the presence of GG (Figure 2a). The expression levels of the drug-metabolizing enzymes CYP3A4, CYP2C9, and CYP2C19 were also significantly increased in the GG group when compared with the control group (Figure 2b). Moreover, the levels of drug transporters markedly increased in the presence of GG (Figure 2c). The protein expression levels of CDX2, villin, DPP4, SGLT1 [29], BCRP, and PEPT1 [30,31] were evaluated with or without GG (Figure 2d,e). Moreover, the percentages of SGLT1-, BCRP-, and PEPT1-positive cells were compared between the control and GG groups via immunofluorescence staining. A quantitative analysis of the proportion of positive cells relative to the control group showed that GG significantly increased protein expression levels (Figure 2e).

### 3.2. Activities of the Drug-Metabolizing Enzymes and Peptide Uptake Transporter

The differentiated cells were positive for the major drug-metabolizing enzymes in the small intestine, namely CYP3A4/5, CYP2C9, and CYP2C19 (Figure 3a–c). Notably, CYP2C19 activity was higher in the GG group than in the control group. CYP3A4/5 activity was significantly inhibited by ketoconazole in all groups. In the small intestine, PEPT1 plays the role of an importer and mediates the uptake of di- and tri-peptides from the lumen into the enterocytes. Therefore, we determined the activity of PEPT1 in differentiated cells using glycylsarcosine, which is the substrate of PEPT1. Glyclysarcosine uptake was higher in the GG group than in the control group. Moreover, the absorption was significantly suppressed in all groups at 4 °C. In the presence of ibuprofen, the uptake was markedly inhibited in the GG group compared with that in the control group (Figure 3d). 

### 3.3. Analyses of Cellular Senescence and Adhesion

Using immunofluorescence, we found that the number of differentiated enterocytes in the GG group was higher than that in the control group (Appendix A). Gellan gum can inhibit cell proliferation by inducing cells to enter the G0 phase [32]. To verify the relationship between the promotion of enterocyte differentiation and the increase in cell number, we analyzed the cell cycle changes in the differentiated cells via flow cytometry. The number of cells in each phase was not significantly different between the GG and control groups (Appendix A). Therefore, we evaluated the number of senescent cells via β-galactosidase staining [33]. The number of β-galactosidase-positive cells was higher in the GG group than in the control group (Figure 4a,b). The gene expression level of p21, a senescence marker [34], was higher in the GG group than in the control group (Figure 4c). Moreover, the gene and protein expression levels of the cell matrix protein integrin α5 were confirmed (Figure 4d,e). In addition, the protein expression level of integrin α5 was significantly higher in the GG group than in the control group (Figure 4f).

## 4. Discussion

According to the FDA, the clinical success rate during novel drug development is less than 10% [35]. The high failure rates correlate with each step of drug development, and the lessons learned from the early stages directly impact the subsequent stages. It is desirable to accurately predict novel drugs. Because the prediction models used for novel drug efficiency evaluations have some disadvantages, it is difficult to precisely assess drug absorption and metabolism in the small intestine. Therefore, human iPS cell-derived enterocytes with sufficient pharmacokinetic functions could be useful during the pre-clinical trial and may help in the accurate evaluation of new drugs.

Mimicking the in vivo environment has been used as a novel approach to maintain or increase the tissue-specific functions of the cells. For example, the differentiation of small intestinal cells can be regulated by coculturing them with neurons, glia, or subepithelial myofibroblasts of the enteric nervous system [36]. Fluid shear stress caused by a micro-device and media perfusion using a pump affect the polarity and metabolism of Caco-2 cells [37]. However, the manufacturing and further application of these devices remain difficult. In this study, we developed a simple method to optimize enterocyte differentiation from iPS cells by adding dietary fibers to mimic the environment of the small intestine. We cultured the cells during differentiation with GG, a soluble dietary fiber. The differentiated enterocytes showed characteristics similar to the small intestine regarding the expression of various genes and proteins. The small intestine contains several drug-metabolizing enzymes, including CYP3A4 (82%), CYP2C9 (14%), and CYP2C19 (2%) [38]. CYP3A4 has an essential role in drug metabolism in the small intestine. To validate the presence of functional drug-metabolizing enzymes in the differentiated enterocytes, we used the proven substrates of each enzyme and measured the metabolite content. All three enzymes were functional. The activity of CYP3A4 was significantly suppressed by ketoconazole, a CYP3A4 inhibitor, in both the control and GG groups. We hypothesized that the differentiated enterocytes expressed drug-metabolizing enzymes. The pharmacokinetic function of PEPT1 was observed in the control and GG groups. Furthermore, compared with the control group, the GG group showed higher PEPT1 activity. Therefore, we used GG to promote the differentiation of human iPS cells into enterocytes. Moreover, we found that GG strengthened the cell matrix adhesion ability of the enterocytes. Increased expression of integrin α5 protects enterocyte detachment from the extracellular matrix by a process called anoikis. Integrin α5β1 protects bone marrow mesenchymal stem cells from anoikis [39]. Furthermore, integrin endosomal signaling correlates with reduced anoikis sensitivity [40]. The findings of this study are consistent with the findings of these reports. We found that the percentage of senescent cells was increased in the presence of GG. Normal cells that exit the cell cycle can become senescent via DNA damage [41], production of inflammatory mediators, or increased reactive oxygen species (ROS). Programmed death and cell proliferation do not occur in senescent cells. Aging leads to few changes in the small intestine; however, pronounced changes occur in the colon [42]. There are no significant changes in the morphology of the small intestine during aging [43]. However, villi height decreases with age [44]. The correlation between intestinal morphology, functions, and aging remains controversial. In our study, we confirmed that enterocytes differentiated from human iPS cells in the presence of GG showed characteristics similar to small intestinal cells. GG helps differentiated enterocytes maintain longevity by inducing senescence.

Previous studies have shown that vascular smooth muscle cells (SMCs) can survive with higher viability in a suspension culture with GG. GG can arrest vascular smooth muscle cell growth by inhibiting their entry into the cell cycle via the induction of p27^Kip1^ [32]. However, p21^Cip1^ expression was reduced in the presence of GG, although the level of p21^Cip1^ was higher in our study. p21^Cip1^ is not only an apoptosis-related protein but also a senescence trigger [34,45]. We hypothesized that the increase in the number of senescent cells was due to an increase in p21^Cip1^. The differences in cell types may have caused the inconsistent results. Additionally, in the previous study, GG played the role of a scaffolding substrate by virtue of its viscosity. However, in our study, the medium with GG showed almost no viscosity. This may be another reason for the different results.

In this study, we proved that the dietary fiber GG promotes the differentiation of human iPS cells into enterocytes with better small intestinal-specific function. This is the first study to demonstrate the usefulness of dietary fibers in cellular differentiation.

## 5. Conclusions

To obtain human iPS cell-derived enterocytes with better functions, we differentiated enterocytes in the presence of GG. The differentiated enterocytes had higher mRNA and protein expression levels of small intestinal markers, drug-metabolizing enzymes, and drug transporters. The activities of CYP3A4/5, CYP2C9, and CYP2C19 were also observed in human iPS cell-derived enterocytes, among which the activity of CYP2C19 was significantly increased. The uptake activity via PEPT1 was higher in the presence of GG. Therefore, GG promotes the differentiation of human iPS cells into enterocytes. In addition, the percentage of senescent cells and integrin α5 protein expression level increased in the presence of GG. Therefore, GG may protect differentiated cells from death and promote cell maturity by increasing cell–ECM adhesion and senescence.

## Figures and Tables

**Figure 1 pharmaceutics-12-00951-f001:**
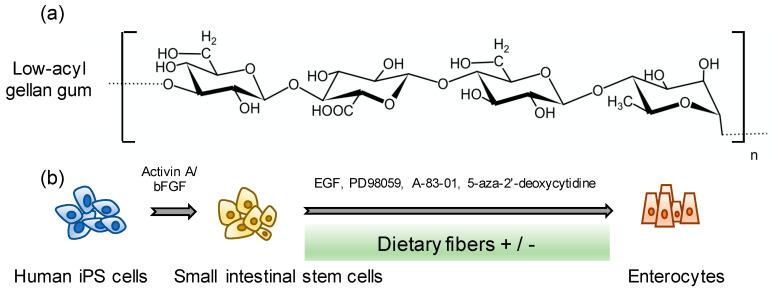
Scheme of human induced pluripotent stem (iPS) cell-derived enterocyte differentiation. (**a**) Structure of low-acyl gellan gum. (**b**) Human iPS cell-derived enterocyte differentiation method.

**Figure 2 pharmaceutics-12-00951-f002:**
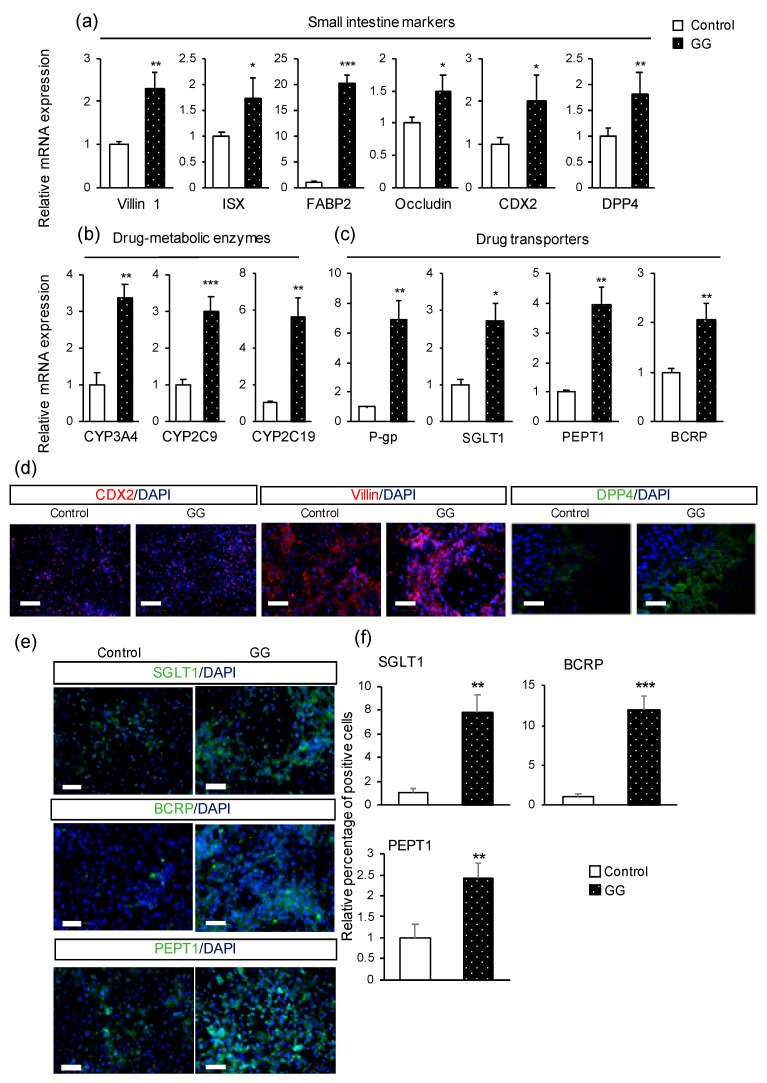
The effect of GG on the differentiation of human iPS cells into enterocytes. (**a**–**c**) Relative gene expression levels of small intestinal markers, drug-metabolizing enzymes, and drug transporter markers. (**d**) Immunofluorescence images of the small intestinal marker caudal-type homeobox 2 (CDX2) and enterocyte markers villin and dipeptidyl peptidase-4 (DPP4). (**e**) Immunofluorescence images of the drug transporter markers sodium-glucose co-transporter 1 (SGLT1), breast cancer resistant protein (BCRP), and peptide transporter 1 (PEPT1). (**f**) Relative percentage of positive cells of drug transporters. Scale bars = 100 μm. All data are presented as mean ± S.D. (*n* = 3). Control = 1. Levels of statistical significance: * *p* < 0.05, ** *p* < 0.01, *** *p* < 0.001.

**Figure 3 pharmaceutics-12-00951-f003:**
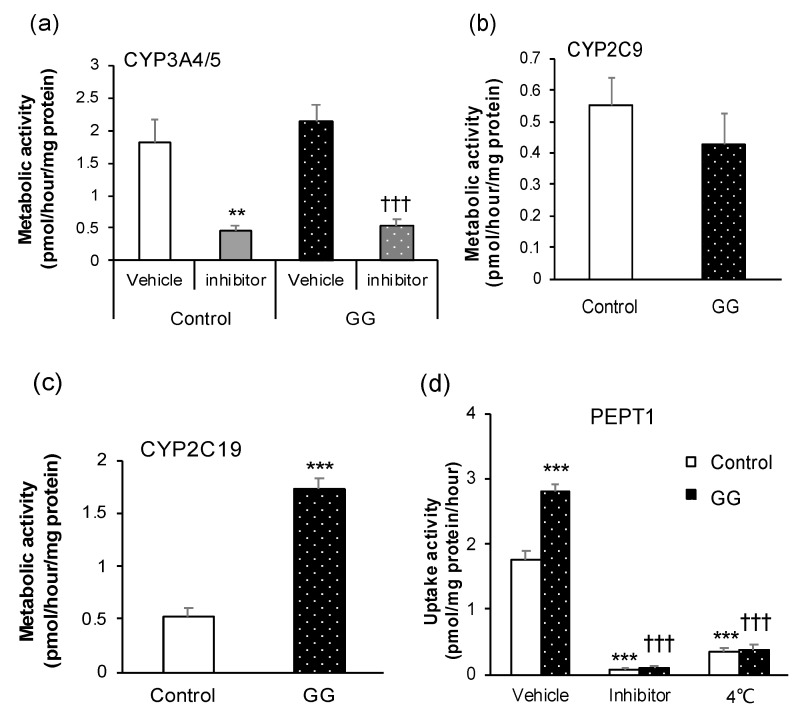
Pharmacokinetic functions in the differentiated enterocytes. (**a**) Cytochrome P450 (CYP) 3A4/5 activity in the presence or absence of 10 μM ketoconazole, a CYP3A4 inhibitor. (**b**,**c**) CYP2C9 and CYP2C19 activities. (**d**) Uptake activity via peptide transporter 1 (PEPT1). Ibuprofen was used as an inhibitor of PEPT1. All data are presented as mean ± S.D. (*n* = 4). Levels of statistical significance compared with the control group (vehicle): ** *p* < 0.01, *** *p* < 0.001. Levels of statistical significance compared with the GG group (vehicle): ††† *p* < 0.001.

**Figure 4 pharmaceutics-12-00951-f004:**
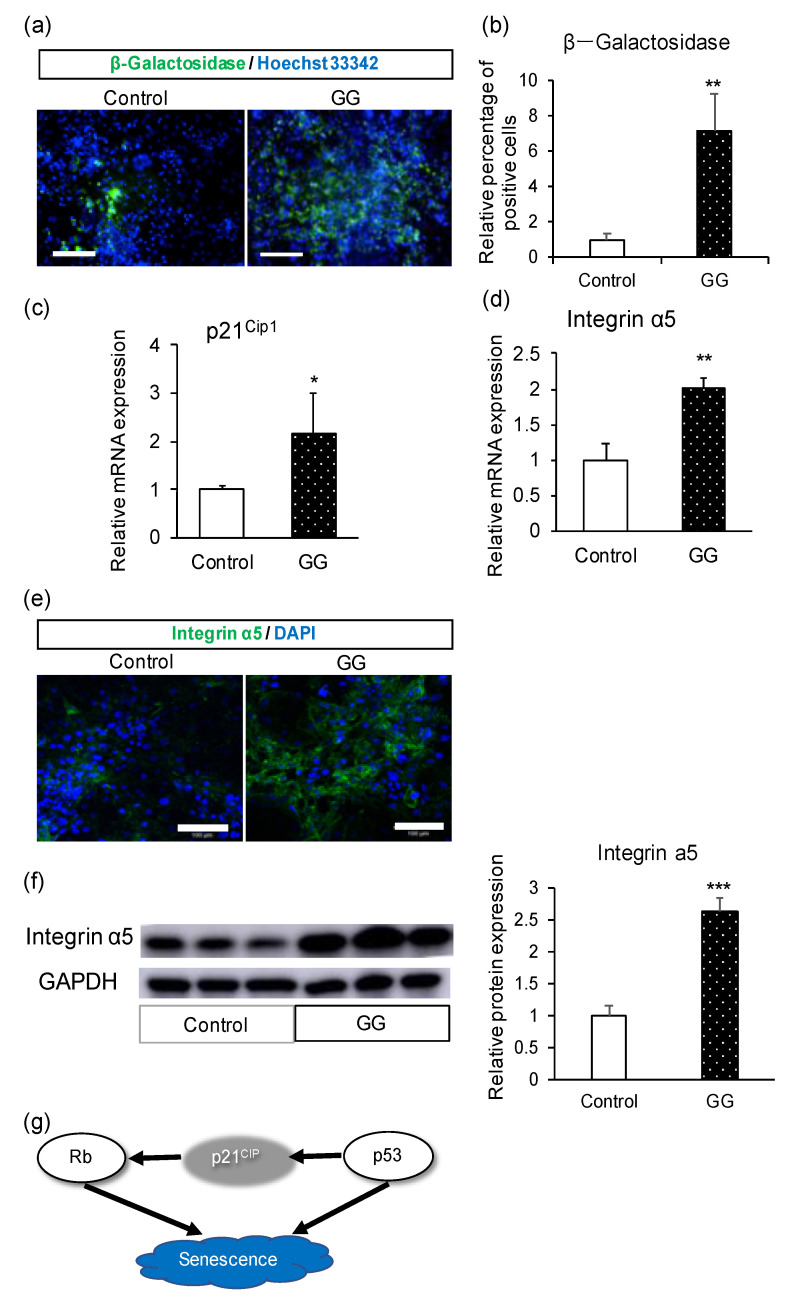
Mechanistic analysis of the effect of GG in enterocyte differentiation. (**a**) Immunofluorescence images of the senescence marker β-galactosidase. (**b**) Relative percentage of β-galactosidase-positive cells. (**c**) Relative mRNA expression levels of p21^Cip1^. (**d**) Relative mRNA expression levels of integrin α5. (**e**) Immunofluorescence images of integrin α5. (**f**) Western blotting analysis of integrin α5 and relative protein expression. (**g**) Proposed scheme showing the relationship between p21^Cip1^ and senescence. Scale bars = 100 μm. All data are presented as mean ± S.D. (*n* = 3). Control = 1. Levels of statistical significance compared with the control group: * *p* < 0.05, ** *p* < 0.01, *** *p* < 0.001.

**Table 1 pharmaceutics-12-00951-t001:** Sequences of the primers used for real-time PCR analysis.

Gene Name	Sense (5′→3′)	Antisense (5′→3′)
Villin1	AGCCAGATCACTGCTGAGGT	TGGACAGGTGTTCCTCCTTC
ISX	CAGGAAGGAAGGAAGAGCAA	TGGGTAGTGGGTAAAGTGGAA
FABP2	TTGGAAGGTAGACCGGAGTG	AGGTCCCCCTGAGTTCAGTT
Occludin	CTGTGTGTTTCCAAGAGAAGTTAC	TGCATCATCAATTTCCTCCTGCAG
CDX2	ACCTGTGCGAGTGGATGC	TCCTTTGCTCTTGCGGTTCT
DPP4	CAAATTGAAGCAGCCAGACA	GGAGTTGGGAGACCCATGTA
CYP3A4	CTGTGTGTTTCCAAGAGAAGTTAC	TGCATCATCAATTTCCTCCTGCAG
CYP2C9	GACATGAACAACCCTCAGGACTTT	TGCTTGTCGTCTCTGTCCCA
CYP2C19	GAACACCAAGAATCGATGGACA	TCAGCAGGAGAAGGAGAGCATA
P-gp	CCCATCATTGCAATAGCAGG	TGTTCAAACTTCTGCTCCTGA
SGLT1	CAACATCGCCTATCCAACCT	TAAACAACCTTCCGGCAATC
PEPT1	CACCTCCTTGAAGAAGATGGCA	GGGAAGACTGGAAGAGTTTTATCG
BCRP	AGATGGGTTTCCAAGCGTTCAT	CCAGTCCCAGTACGACTGTGACA
Integrin α5	TGCAGTGTGAGGCTGTGTACA	GTGGCCACCTGACGCTCT
p21^Cip1^	TGTCCGTCAGAACCCATGC	AAAGTCGAAGTTCCATCGCTC
HPRT	CTTTGCTTTCCTTGGTCAGG	TCAAGGGCATATCCTACAACA

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
