# Peer review of "Gellan Gum Promotes the Differentiation of Enterocytes from Human Induced Pluripotent Stem Cells"

_pharmaceutics, 2020, doi:10.3390/pharmaceutics12100951_

Round 1
Reviewer 1 Report
Review of Manuscript ID: 948925 entitled ‘Gellan gum promotes the differentiation of enterocytes from human induced pluripotent stem cells’ by Shimeng Qiu and colleagues.
The above titled manuscript by Shimeng Qiu and co-workers is a research article on the use of Gellam Gum to try to improve the differentiation efficiency of human iPS cells into enterocytes. The authors used gellan gum, a soluble dietary fiber, to optimize hiPSE differentiation. The authors observed that hiPSEs cocultured with GG had a significantly higher expression of small intestine- and pharmacokinetics-related genes and proteins. In addition, the activities of drug metabolizing enzymes including cytochrome P450 2C19, and peptide transporter 1 were significantly increased in the GG treatment group than in the control group. Overall, the authors concluded that GG could improve the differentiation efficiency of human iPS cells to enterocytes and increase intestinal maturation by extending the life span of hiPSEs.
This is a well-written manuscript, with well-designed experiments. However the manuscript lacks a very important step as described below:
Specific Points
- The most important step missing from the manuscript is the characterisation of the iPS used in the study. The authors referred the reader to Reference 13, but a closer look also show no characterisation of pluripotent stem cells was done in that reference. It is important that IPSs and other stem cells are characterised before being used in experiments- show expression of markers of pluripotent cells as well as being able to differentiate into the three lineages as done in PMID: 27527147 PMCID: PMC5000657 DOI: 10.3390/ijms17081259.
- In addition to the above, all research involving human cells, must have Ethical Approval by the relevant authorities. None is given in the manuscript.
- Figure 4 F – GAPDH blots are normally put under those of the gene of interest and not above.
- Figure 4 G – What is shown is a ‘Proposed Scheme’ and is still to be verified and validated, including gene knockdowns and overexpression experiments.
Reviewer 2 Report
The authors have investigated the effect of gellan gum on the differentiation of human induced pluripotent stem cells into ADME relevant intestinal enterocytes.
The work is relevant as there are several obvious shortcomings with current in vitro cell models of intestinal permeability related to morphological and functional differences compared to the human small intestine. I am not qualified to review the method by which the stem cells were grown and differentiated. However, I have comments relating to the results, discussion and conclusions in general based on my drug delivery and ADME experience.
My primary concern with this manuscript is that only RNA/protein expression is reported, whereas function is at no time investigated. For instance, I am very interested to see the permeability and metabolic characteristics of a set of models drugs in this model, compared to accept ones such as caco2. I am also lacking a mechanistic explanation for how dietary fibres influence enterocyte differentiation.
Specific comments
Line 36. The manuscript is in high need of a language revision by a specialized English proofreader. This is one (of many) sentences that makes little sense “For a drug to be absorbed and metabolized by the small intestine and liver [3-5], it is important to accurately evaluate 38 the factors for predicting the effectiveness and safety of the novel drug.”
Line 41. “First, the expression patterns of drug transporters in Caco-2 cells are different from those in the normal human small intestine [8-10].” This paper discusses and focus a lot on transporters. Especially on transporters that have none, or insignificant, impact on drug disposition, such as Pept1 and SGLT1. In what way does this say anything regarding the ADME utility of this in vitro system?
Line 52. “Several studies have demonstrated that some small molecular compounds could improve the differentiation of human iPS cells into enterocytes [13-16]. However, such enterocytes poorly function compared with the small intestine cells in the human body.” In what way do they “poorly function”? And how do you know that your system is not poorly functioning, as you have not investigated its function?
Line 55. “Therefore, we focused on generating enterocytes with higher pharmacokinetic properties from human iPS cells.” What is “higher pharmacokinetic properties”?
Line 62. “We produced human iPS cell-derived enterocytes (hiPSEs), with improved pharmacokinetic abilities by adding GG during differentiation and identified the mechanism of GG-mediated differentiation.” Where have you identified the mechanisms of GG-mediated differentiation? I do not see that.
Line 64. “Our results suggest that GG extends the life span of the mature cells and maintains cellular function.” First of all, one does not present results and conlcusions in the introduction of a scientific article. Second, in what way does your result suggest that GG extends the life span of mature cells?
Figure S2. Why is there a high effect of GG at 0.01%, but not at the higher 0.015%? Why where these concentrations investigated? Why did you investigate two concentrations that were so similar?
Figure 2 a-b. There seems to be no difference in the metabolic capacity of CYP3A4 between control and GG. This is the most important CYP enzyme of the intestine. Doesn’t this indicates that GG has limited impact of the usefulness of this system for increasing its relevance for metabolic preclinical studies.
Figure 2d. Pept1 is not important for drug disposition and absorption. Most drugs are absorbed by the passive route. What is the utility of this model for studying that?
How does this model compare to caco 2 and/or in vivo with regards to transporter and metabolic enzyme expression.
The discussion is in need of sections.
Round 2
Reviewer 2 Report
OK